# Diamagnetism of Bulk Graphite Revised

**Bogdan Semenenko and Pablo D. Esquinazi** \*

Division of Superconductivity and Magnetism, Felix-Bloch-Institute for Solid State Physics, Faculty of Physics and Earth Sciences, University of Leipzig, Linnéstraße 5, 04103 Leipzig, Germany; semenenko@studserv.uni-leipzig.de

\* Correspondence: esquin@physik.uni-leipzig.de; Tel.: +49-341-97-32-751

**Abstract:** Recently published structural analysis and galvanomagnetic studies of a large number of different bulk and mesoscopic graphite samples of high quality and purity reveal that the common picture assuming graphite samples as a semimetal with a homogeneous carrier density of conduction electrons is misleading. These new studies indicate that the main electrical conduction path occurs within 2D interfaces embedded in semiconducting Bernal and/or rhombohedral stacking regions. This new knowledge incites us to revise experimentally and theoretically the diamagnetism of graphite samples. We found that the *c*-axis susceptibility of highly pure oriented graphite samples is not really constant, but can vary several tens of percent for bulk samples with thickness $t \gtrsim 30$ μm, whereas by a much larger factor for samples with a smaller thickness. The observed decrease of the susceptibility with sample thickness qualitatively resembles the one reported for the electrical conductivity and indicates that the main part of the *c*-axis diamagnetic signal is not intrinsic to the ideal graphite structure, but it is due to the highly conducting 2D interfaces. The interpretation of the main diamagnetic signal of graphite agrees with the reported description of its galvanomagnetic properties and provides a hint to understand some magnetic peculiarities of thin graphite samples.

**Keywords:** graphite; diamagnetism; thickness dependence; susceptibility; interfaces; conductivity; graphene

---

## 1. Introduction

The *c*-axis diamagnetic susceptibility of graphite is very large and anisotropic [1–3]. According to the literature of the last 50 years, there is consent to interpret this large diamagnetism as due to the Landau diamagnetic contribution of a certain density of free conduction electrons within the graphene planes of graphite. The relatively low density of conduction electrons in graphite arises from the overlap of the $2p_z$ electronic orbitals, normal to the graphene planes; whereas the overlap between those orbitals from the carbon atoms at neighboring graphene layers, in both Bernal and rhombohedral stacking orders, remains very weak, i.e., van der Waals coupling, as the huge anisotropy of the resistivity and magnetization indicates.

The calculations of the conduction-electrons magnetic susceptibility have been done in the past taking into account an electronic band structure inferred from electric transport and magnetic measurements [4–6]; in particular, using the quantum oscillations in the electrical resistance, the Hall effect, and magnetization, i.e., the Shubnikov–de Haas (SdH) and de Haas–van Alphen effects. All theoretical models, as well as the interpretation of the measured diamagnetism of graphite in those publications assumed that high quality and pure graphite samples are homogeneous, structurally, as well as electronically. All free parameters of the band structure models were obtained from a comparison with experimental data of different graphite samples [7]. Nowadays, one may doubt the accuracy of the used models in particular because no electron-electron or spin-orbit coupling

interactions were included explicitly, plus the difficulties those calculations have in modeling the van der Waals interactions between the graphene layers.

The main problem those models and interpretation have, however, is directly related to the misleading assumption that the experimental magnetic and electrical data correspond to homogeneous bulk graphite samples. First experimental hints at odds with this assumption were obtained from the magnetic field dependence of the Hall coefficient of kish graphite samples of different thicknesses [8,9]; namely, the amplitude of the SdH oscillations decreases the smaller the thickness of the samples. For example, for a sample with a thickness of 18 nm (which corresponds to a stacking of more than 50 graphene layers), one can barely recognize the field oscillations in the Hall coefficient, in clear contrast to thicker flakes; for a review and discussion on these and other experimental results on this issue, see [10]. Moreover, a nonlinear increase of the resistance of graphite samples of a smaller thickness was reported [11], which can be described as an anomalous increase in the estimated absolute resistivity [12] the thinner the sample. Surprisingly, none of those studies [8,9,11] tried to correlate the obtained results with the internal structure of the samples.

Experimental evidence obtained in graphite bulk samples and thin flakes over the last 10 years reveals that the observed thickness dependence of the magneto-electric properties of graphite has its origin in the internal microstructure of the samples [10,12–14]. In particular, scanning transmission electron microscopy (STEM) studies (as an example, see Figure 1) reveal the existence of two-dimensional (2D) interfaces between regions with different stacking order and/or between regions with similar stacking order, but twisted around a common $c$-axis [10]. The presence of two stacking orders, identified as the majority phase, called Bernal (ABABA...) and the rhombohedral (ABCABCA...) stacking order, has been measured by high resolution X-rays diffraction (XRD) [15] of different samples including natural graphite crystals, highly oriented pyrolytic graphite (HOPG), or kish graphite [10,16], in agreement with previous reports [7,17]. It is important to note that these two stacking orders are not semimetals, but semiconductors with energy gaps of $38 \pm 8$ meV (Bernal) and $110 \pm 20$ meV (rhombohedral), obtained from the fits to the temperature dependence of the resistance between 2 K–1100 K of a large number of different samples of different origins [13].

The galvanomagnetic results obtained for samples of different thickness indicate that a highly conducting path in graphite samples is localized at the 2D interfaces [12–14,18], which upon twist angle [10] or stacking order [19,20], can even show 2D superconducting properties [15,21–23]. The origin of the SdH oscillations in the electrical resistance is related to the 2D interfaces, as recent detailed electrical measurements clearly revealed [14]. All these recent results motivated us to study more carefully the magnetization of graphite samples with smaller thickness than the usually reported samples in the literature. Since the two stacking order phases are semiconducting, it is clear that the diamagnetic signal of graphite cannot be intrinsic to the graphite structure, otherwise we would have large changes of the $c$-axis susceptibility with temperature, which is not the case [7]. If the large diamagnetic moment measured in large bulk samples of graphite is mainly due to the highly conducting interfaces, taking into account STEM studies (see Section 2), we expect then to observe a non-systematic variation of the diamagnetic response in bulk samples, even if the samples have a similar volume and are cut from the same sample. Furthermore, we would expect to see a decrease in the absolute value of the diamagnetic susceptibility the smaller the interface number, i.e., the smaller the sample thickness.

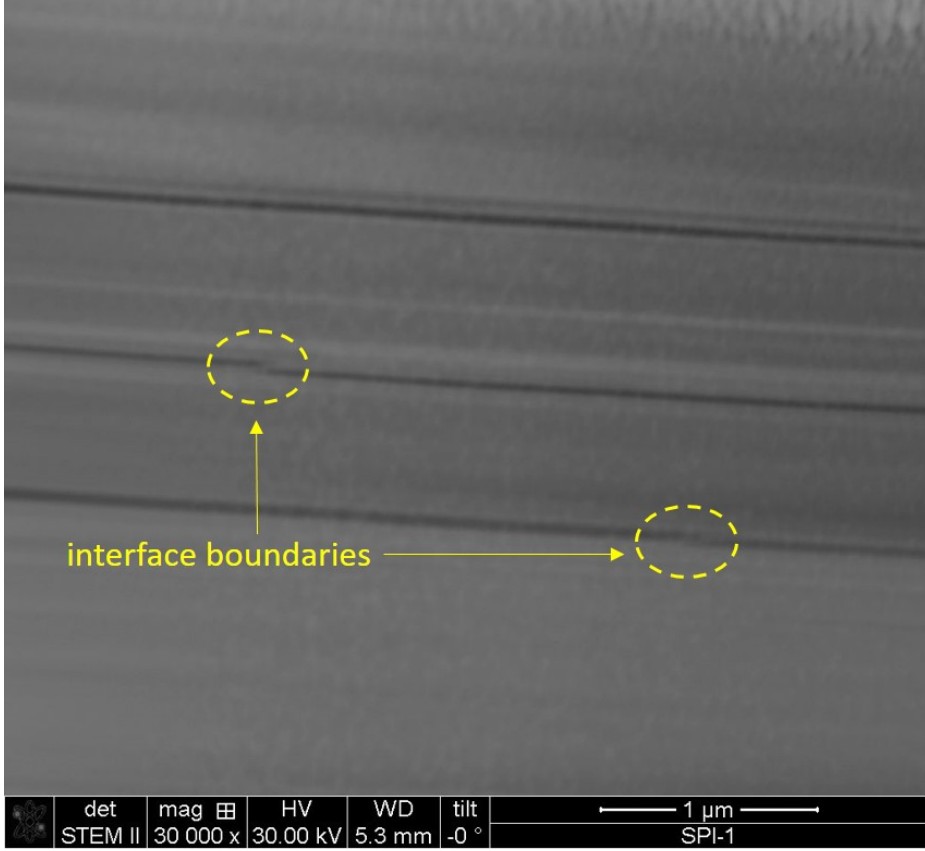

**Figure 1.** STEM picture of a commercial HOPG (SPI) graphite lamella (∼400 nm thick) cut from a bulk sample measured also in this work. The electron beam is applied parallel to the graphene layers and normal to the main *c*-axis of the graphite structure. The different grey colors indicate crystalline regions with different stacking orders (Bernal or rhombohedral) or regions twisted around the *c*-axis by a certain angle with respect to the neighboring regions. Well-defined two-dimensional interfaces are located between regions with different grey colors at the middle of the picture, which includes also interface boundaries (yellow ellipsoids). More STEM pictures (also with much higher resolution) obtained from different commercial HOPG samples and natural graphite can be seen in [10]; for more details on the samples see Section 5.1 below and Section 3 in the Supplementary Information.

In clear contrast to the technical requirements needed to study the electrical resistance of graphite samples with a thickness down to a single graphene layer, the measurement of the diamagnetic moment of thin graphite samples with a commercial SQUID magnetometer is difficult or even impracticable. For example, assuming one wants to measure the diamagnetic *c*-axis magnetic moment of a graphite flake of thickness × width × length equal to 1 µm × 0.2 mm × 0.2 mm with an expected diamagnetic *c*-axis susceptibility $\chi \sim -2 \times 10^{-5}$ emu/g Oe, the expected diamagnetic moment at a field of $10^4$ Oe applied parallel to the *c*-axis would be $m \lesssim -2 \times 10^{-8}$ emu, a value of the order of the error of commercial SQUIDs nowadays. The expected small magnetic moment added to a not-easy handling of such small thin flakes for that kind of measurements (without large backgrounds from substrates, etc.) put already hard restrictions on performing such magnetization measurements.

In this work and besides the SQUID, we have used a torque magnetometer that allowed us to measure with high resolution the susceptibility of well-ordered graphite flakes with a thickness of ∼1 µm and larger. The obtained results of the magnetic moment of highly oriented samples of different sources and with both magnetometers show that the absolute value of the diamagnetic susceptibility decreases the smaller the sample thickness. Our results can be considered as a first experimental hint that agrees qualitatively with the thickness dependence of the conductivity of similar samples.

Our results suggest that the largest contribution to the diamagnetic susceptibility measured in bulk samples is not intrinsic to the ideal graphite structure.

The manuscript is organized as follows: In the next section, we discuss the density of interfaces following the information of an STEM picture obtained from one of the measured samples. Section 3 is divided into three more sections, where we discuss: (A) the temperature dependence of the diamagnetism of graphite samples, (B) the angle dependence of the torque magnetometer, and (C) the thickness dependence of the diamagnetic susceptibility. In Section 4, we discuss the results and propose a simple model to understand at least semiquantitatively the thickness dependence of the diamagnetic susceptibility of graphite samples. Section 5 describes the characteristics of the selected samples and the details of the magnetometers used. The conclusion is given in Section 6.

## 2. Internal Structure of Graphite Samples

The 2D interfaces appear between crystalline regions with different stacking orders or twisted by a certain angle around the common *c*-axis. Whereas ideal Bernal and rhombohedral stacking orders are low-energy gap semiconductors [13,18,24,25], twisted crystalline graphite regions reveal angle-dependent moire patterns at their interface with a much higher and position-dependent electronic density of states, as scanning tunneling microscopy revealed [10,26–29]. The 2D interfaces are located at the boundaries of the regions with different grey colors in Figure 1. This STEM picture tells us that: (a) The density of interfaces is not homogeneous within the observed region of $\sim$3 μm parallel to the *c*-axis. (b) We can recognize clearly only some of the interfaces because of the relatively large thickness of $\sim$400 nm of the lamella. Note that some of the 2D interfaces from in-depth regions of the lamella do not appear with clear boundaries in the STEM picture. (c) The length of the interfaces is in general much less than the length of the graphene planes due to grain boundaries. Two regions with cut and shifted interfaces are indicated by the yellow ellipsoids in Figure 1. This means that only due to the internal microstructure of the graphite samples, the effective weight to the total diamagnetic response of each single interface we recognize in the rather small part of a sample through the STEM picture is in general less than one. In other words, we expect that for mesoscopic and macroscopic graphite samples, the ratio between the effective number of interfaces $N_{int}$, contributing similarly in shielding the applied field within a region of $N$ graphene layers, can have non-integer values. Therefore, the ratio $N_{int}/N$ can be considered as an effective parameter in the model we present in Section 4. (d) Which of the observed interfaces provides the highest diamagnetic response, i.e., magnetic field shielding, remains still unknown. It may be that most of the interfaces between twisted regions react similarly under a magnetic field due to the existence of a similar high density of states at the hexagonal paths observed in the moire patters with different diameters (see [10] and the references therein).

We may conclude that neither the area, nor the density, nor the electronic characteristics of the interfaces are homogeneously distributed within each bulk sample, making cumbersome the interpretation of different properties that depend on the response of these interfaces. For example, if we measured the diamagnetic response of the lamella of Figure 1 as it is, and after removing the interface-free region at the bottom, we would calculate for the lamella with less mass (but with the same amount of interfaces) an enhanced diamagnetic susceptibility with respect to the original sample before. In other words, a normalization of the measured magnetic moment by the total sample mass is, strictly speaking, incorrect.

We note that in general, the diamagnetic *c*-axis magnetization of different graphite samples is not straightforward to understand, even qualitatively. For example, early reports showed a non-monotonous behavior on the degree of graphitization, and its absolute value is $\sim$30% smaller for the highest oriented than for less oriented samples [30]. We believe that at least part of this behavior is related to internal interfaces.

In order to fix the ideas and provide a semiquantitative estimate of the *c*-axis susceptibility due to the internal interfaces, from Figure 1 and other STEM studies, we estimate between $\sim$16 and $\sim$20

interfaces in ∼3 μm parallel to the *c*-axis. Taking into account the distance between graphene planes in graphite, the interface density would be of the order of $N_{int}/N \sim (1.6\ldots2) \times 10^{-3}$. If the main contribution to the total diamagnetic susceptibility of graphite were directly proportional to this ratio, as our estimates suggest (see Section 4), one would expect a decrease of this effective ratio the thinner the sample, below a certain thickness, which depends on the internal structure of the graphite sample.

## 3. Results

(A) Temperature dependence of the diamagnetism of graphite: The temperature dependence of the susceptibility of HOPG samples was obtained using samples of thickness 201 μm and 27 μm. A magnetic field of 10 kOe parallel to the *c*-axis was applied at 300 K, and the samples were cooled to 5 K in the sweep mode with a 2 K/min rate. The results are shown in Figure 2.

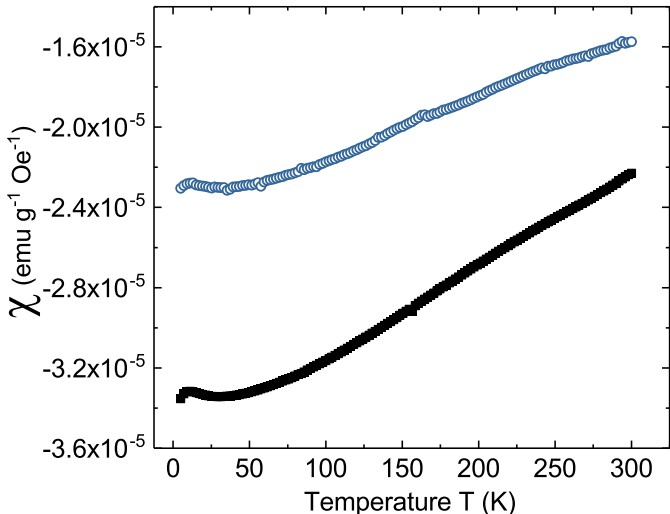

**Figure 2.** Temperature dependence of the mass susceptibility of two HOPG samples from the same source with thicknesses of 201 μm (bottom curve) and 27 μm (upper curve) measured with an SQUID magnetometer.

As already reported in the literature, the change of the diamagnetic *c*-axis susceptibility with temperature is rather weak, having a maximal diamagnetic response at ∼50 K with a slight increase at lower temperatures, as Figure 2 shows. The presented result of the bulk sample is similar to published results (see, e.g., Figure 6 in [31]). The results shown in Figure 2 indicate that the absolute value of the magnetic susceptibility of the thinner sample was ≃25% smaller than samples with a thickness of 30.6 μm or 201 μm at 300 K. The samples having similar areas and prepared from the same bulk sample, this difference is not due to an error in the measurement or because the quality of the sample has been changed through handling. These variations of the susceptibility for similar samples of different thicknesses already suggest a non-intrinsic origin of the main diamagnetic signal.

(B) Angle dependence of the torque: The torque magnetometer can be used to obtain the magnetic moment for one field direction if the sample is strongly magnetically anisotropic, i.e., $\chi_{\parallel} \gg \chi_{\perp}$, where the two susceptibilities mean for fields parallel and perpendicular to the *c*-axis. According to published results, the ratio between the two susceptibilities is $\chi_{\parallel}/\chi_{\perp} \gtrsim 10$ [1,2,30,32]. This means that the torque signal is basically given by the magnetic moment component parallel to the *c*-axis.

Figure 3 shows as example of the angular dependence of the measured torque signal under a constant magnetic field of 2 kOe at room temperature performed on two different pieces of a natural graphite sample of thicknesses 1.2 μm and 6.9 μm. The measurements were done in the two field sweep directions, and as can be seen in the figure, a good reproducibility was achieved with negligible

hysteresis. The magnetic moment obtained from the torque signal depends on the angle $\theta$ between the applied magnetic field $H$ and the *c*-axis of the samples, as shown in Figure 3.

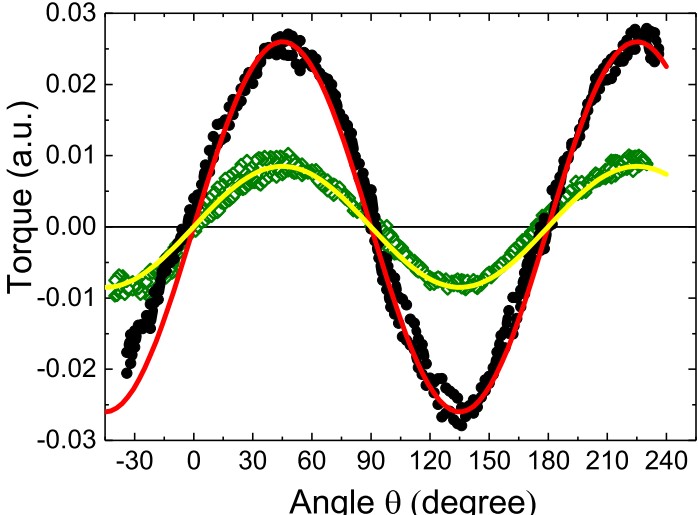

**Figure 3.** Torque signal of two natural graphite flakes vs. angle $\theta$ between the applied field $H = 2$ kOe and the *c*-axis of the graphite structure. The thicknesses of the samples were 1.2 μm ($\Diamond$) and 6.9 μm ($\bullet$). The lines are fits to a $\sin(2\theta)$ function. The magnetic moment of the thinnest sample is of the order of $\sim -2 \times 10^{-9}$ emu.

(C) Thickness dependence of the diamagnetic susceptibility: Figure 4 shows the results of the *c*-axis susceptibility of all graphite samples we have measured at 300 K as a function of their thickness and with the two experimental methods. In the same figure, we included the thickness dependence of the conductivity of graphite obtained at the same temperature (right *y*-axis) taken from [13,33]. Our data basically agree with the published susceptibility data of highly ordered samples of thickness of the order or larger than 100 μm. However, note the variations of $\pm \sim 25\%$ (vertical bar in the upper right in the figure) even for similar samples. We observe that when the thickness of the samples was less than $\sim 50$ μm, dimensional effects began to appear in the electrical, as well as in the magnetic susceptibility of graphite. For $t < 50$ μm, the experimental trend suggests a change of a factor of ten in the susceptibility within a change of $\sim 3$ orders of magnitude in thickness. This roughly means a change of $\sim 30\%$ in one decade of thickness. Note that all measured susceptibility points had a constant, practically thickness independent background contribution coming from the rest of the graphene (mainly Bernal stacking) layers; see Section 4. In Section 1 of the Supplementary Information, we also show the behavior of the resistivity at 4 K as a function of the sample thickness. In Section 2 of the Supplementary Information, we demonstrate that the observed decrease of the measured susceptibility with thickness is not related to a decrease of the total or lateral sample area.

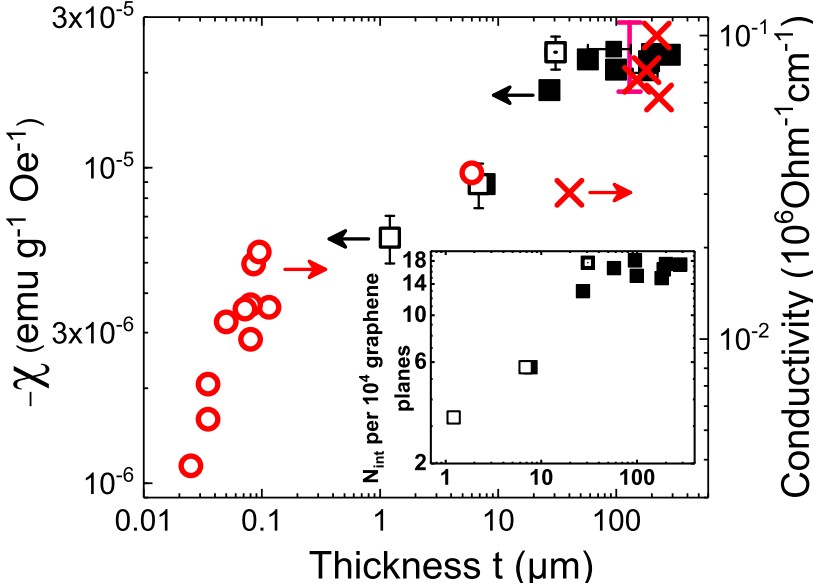

**Figure 4.** *c*-axis susceptibility of different graphite samples vs. their thickness at 300 K measured by a torquemeter (□) and an SQUID magnetometer (■). The samples were obtained from pre-characterized bulk natural graphite [15], as well as HOPG samples of Grade ZYA [13,15,34]. The HOPG sample with a thickness 30.6 μm was measured with the SQUID, as well as with the torque magnetometer (⊡, see Section 4 in the Supplementary Information for details on the calibration of the torque magnetometers). The vertical error bar at the top right indicates the values at 300 K reported for differently-oriented samples in the literature [30,31,35–39]. Right *y*-axis: thickness dependence of the conductivity of several bulk and thin graphite samples at 300 K from (○) [13] and (×) [33], calculated using the given geometry in those publications. The inset shows the estimated effective number of interfaces (per $10^4$ graphene layers) that contribute to the diamagnetic signal as a function of the sample thickness, obtained using Equation (1).

The observed behavior strongly suggests that the observed internal microstructure of the graphite samples plays a major role in those properties. The transport [10,13,14] and structural studies [15] indicate that graphite samples have to be considered, electronically speaking, not as homogeneous, but as a heterostructures. In other words, each sample with thickness $\gtrsim$ 20 nm may start showing three contributions to the conductivity, as well as to the susceptibility: two semiconducting-like phases with Bernal and rhombohedral stacking orders and some of the interfaces between the semiconducting phases or twisted by a certain angle within the same stacking order [10], with a metallic and/or superconducting character [10,13,14,22,23,40]. These three contributions are conducted in parallel when the current is applied parallel to the graphite layers. Decreasing therefore the thickness of the sample below a certain thickness, the diamagnetic contribution to the magnetization, given in first approximation by the simple addition of the three magnetic contributions in series, would decrease because the largest contribution proportional to the diamagnetic response of the interfaces would also decrease.

## 4. Discussion

The decrease of the absolute value of the susceptibility decreasing the thickness of the ordered graphite samples does not appear to be related to extra defects one may introduce through handling or sample preparation (see also Section 3 in the Supplementary Information). Neither the error in the misalignment between the *c*-axis and the applied field direction, nor the error in the sample dimensions, nor mass can provide the changes of ±25% observed in thicker samples or a factor of five in the susceptibility of thinner samples; see Figure 4. The influence of localized spins at the graphene edges (or the presence of nitrogen impurity atoms) [41] on the total susceptibility of our graphite

samples should be negligible, since the lateral surface of the samples is much smaller than their volume, and the magnitude of the total susceptibility does not correlate with the lateral surface of the samples (see Section 2 in the Supplementary Information). Moreover, our magnetization measurements were carried out at room temperature, which strongly diminishes any ferromagnetism or paramagnetism influence due to edge states or other possible localized magnetic moments. It should be clear that due to the heterostructure of most of the graphite samples, i.e., the existence of highly conducting (or superconducting) interfaces and the semiconducting character of the two stacking orders, the magnetic properties of graphite cannot be explained with the models proposed in the literature [4–6], so new approaches are needed.

To understand at least semiquantitatively the observed behavior, we take a simple approach, the main aim of which is to estimate the order of magnitude of the total measured magnetic moment of the sample $m$ assuming that it is due to the direct sum of the independent moments from the 2D interfaces $m_{int}$, the Bernal ($m_B$), and the rhombohedral ($m_{RH}$) stacking orders as: $m = m_{int} + m_B + m_{RH}$. The estimates done below provide the order of magnitude and an explanation for the thickness dependence of the total susceptibility.

We note that there are neither high-resolution band structure measurements, nor calculations, especially for the Bernal stacking orders that provide the obtained small energy gap. This is necessary to get the dispersion relation, the Fermi velocity, and the effective carrier mass at low enough energy, which eventually can be used to estimate the susceptibility of each of the stacking orders. Therefore, we shall assume that each of those contributions is given by the 2D susceptibility of the graphene layers in each stacking order, similarly for the 2D highly conducting interfaces, multiplied by their corresponding densities. The measured $c$-axis susceptibility $\chi$, with the magnetic field $H$ applied normal to the graphene and interface planes, is estimated as:

$$\chi = a(\chi_{int}\frac{N_{int}}{N} + \chi_B\frac{N_B}{N} + \chi_{RH}\frac{N_{RH}}{N}), \tag{1}$$

where the parameters $N_i (i = int, B, RH)$ refer to the effective number of interfaces and the number of graphene layers that belong to the Bernal and rhombohedral phases in a given sample; $N$ is the total number of layers in a given sample of thickness $t$. The prefactor $a$ in Equation (1) is a normalization factor inversely proportional to the 2D mass density. It can be roughly calculated or obtained directly from a comparison with the measured $\chi$ at large enough thickness. The paramagnetic Pauli susceptibility of free carriers may compensate the diamagnetic contribution of the core susceptibility [42]. Furthermore, those contributions are not expected to change with sample size. Therefore, we do not take them into account to estimate the total diamagnetic susceptibility of graphite.

To estimate the $c$-axis magnetic susceptibility of the two semiconducting phases with energy gaps $\sim$38 meV and $\sim$110 meV, we take the formula for the orbital diamagnetic susceptibility of a graphene layer with a band gap $\Delta$ given by (in cgs units) [43]:

$$\chi_\Delta = -g_v g_s \frac{e^2 v_F^2}{6\pi c^2}\frac{1}{2\Delta}, \tag{2}$$

where $g_s = 2$ and $g_v = 2$ represent the degrees of freedom associated with spin and valley, respectively, and $c$ the light velocity. We estimate the electron velocity $v_F$ for a 2D electron system as:

$$v_F = \frac{\hbar\sqrt{2\pi n}}{m^\star}, \tag{3}$$

where $m^\star$ is the effective mass of electrons. From experiments, one can obtain a carrier concentration at room temperature $n \sim 10^{10}$ cm$^{-2}$ [10,18,22]. With this, we estimate a carrier velocity $v_F \sim 3 \times 10^5$ cm/s assuming that for these semiconducting phases, the effective mass is equal to the free electron mass with a quadratic dispersion relation. Future measurements should clarify the value of the effective mass at least for the majority Bernal phase (without interfaces). The 2D diamagnetic susceptibilities

are $\chi_B \sim -4 \times 10^{-17}$ emu/cm$^2$Oe and $\chi_{RH} \sim -10^{-17}$ emu/cm$^2$Oe, where we used the same $n$ for both stacking orders. We expect that $\chi_{RH}$ should be even smaller than the estimate above.

The 2D diamagnetic susceptibility of the internal interfaces is obtained from the expression for the Landau diamagnetism of a 2D gas of free electrons, taking from experiments the effective mass $m^\star \sim 0.05\ m_e$ [44,45] ($m_e$ is the free electron mass). Note that the SdH oscillations in the magnetoresistance are related to the carriers at the interfaces and not to the semiconducting layers. This is the reason why the SdH oscillation amplitude vanishes the smaller the sample thickness [8–10]. This is obvious because the thinnest, semiconducting samples without interfaces should have a negligible amount of conduction electrons at low enough temperatures [10,14].

The Landau diamagnetic susceptibility of the interfaces is therefore:

$$\chi_{int} = -\frac{e^2}{12\pi m^\star c^2} \sim -2 \times 10^{-13}\ \text{emu/cm}^2\text{Oe}\,. \tag{4}$$

From STEM studies, we know that the ratio of the number of interfaces and of the graphene planes with one or other stacking orders depend on the sample and on the position on the same sample; see, e.g., [10,15]. Furthermore, although each interface is parallel to the graphene layers, they do not cover all the mesoscopic sample area; they are limited at least within the single crystalline regions with a length and width within a range of $\sim 1 \ldots 20$ µm, as electron backscattering diffraction pictures indicate [46,47]. This means that a single interface covers in general an area much smaller than the one of the graphene planes. As the thickness of the sample decreases, the probability of having a similar diamagnetic response due to the effective distribution of 2D interfaces also decreases. In other words, the relative weight of each single interface is in general less than one compare to the graphene layers that cover the whole sample area.

To estimate the total susceptibility vs. thickness, with the knowledge of the typical thickness of the Bernal and rhombohedral crystalline regions and the number of embedded interfaces obtained from the STEM images, one can roughly estimate an effective ratio of the number of interfaces in a given sample to the number of graphene layers $\frac{N_{int}}{N}$. For example and to fix the ideas, the following trends are rather general $\frac{N_{int}}{N}(t \lesssim 50\ \text{nm}) \to 0$ and $\frac{N_{int}}{N}(t > 50\ \text{µm}) \sim 1.6 \times 10^{-3}$. The range of the ratio of graphene planes of the Bernal phase is $\frac{N_B}{N} \sim 0.8 \ldots 1$ and of the rhombohedral $\frac{N_{RH}}{N} \sim 0.2 \ldots 0$.

This estimate indicates that in real ordered graphite samples with thickness $t \gtrsim 30$ µm and for a typical ratio of the number of interfaces, the total susceptibility is given mainly by the conducting interfaces. Moreover and in first approximation, the rhombohedral contribution to the total susceptibility can be neglected. With the estimates given above, using the measured value of the susceptibility $\chi(t > 50\ \text{µm}) \sim -2 \times 10^{-5}$ emu/gOe, we obtain $a \sim 6 \times 10^{10}$ cm$^2$/g. From the experimental data and using Equation (1), we estimate the ratio $N_{int}/N$ vs. the sample thickness, shown as the inset in Figure 4.

Future experiments should try to measure the susceptibility of graphite samples with a smaller thickness down to $\sim 10$ nm and area $\lesssim 1$ µm$^2$, because the probability to have the contribution of interfaces is evidently smaller. If these measurements are achieved successfully, one can obtain the susceptibility of the Bernal phase and compare with the theoretical model. This susceptibility represents a rather constant background of the experimental points shown in Figure 4. However, the technical difficulties to measure such small samples with the systems available nowadays are difficult to overwhelm.

Finally, we would like to pay attention to an actually common observation in the laboratories, when one tries to leave a thin graphite flake completely or partially levitating under an inhomogeneous magnetic field from a permanent magnet. We observed that not all thin flakes react similarly to the same field distribution, even when they have similar mass and shape. Several of them do not even react at all to the magnetic field, independently of how small their mass is. This simple observation already indicates that our simple assumption of a homogeneous diamagnetic response of graphite

samples cannot be correct. Within our interpretation, these observations can simply indicate that the selected thin samples have different amounts of conducting interfaces.

Furthermore, an interesting observation was included a year ago on YouTube; namely, large, thin, and flat pieces of pyrolytic graphite that levitate on a north-south chessboard grid of neodymium magnets can be moved, tilted, or shifted by the application of a strong enough laser beam [48]. This observation can be related to an increase in the temperature of the interfaces; the provoked movement is due to a local decrease of the diamagnetic response in the sample. It would be interesting to measure the temperature of the sample during heating and observe changes in its levitation behavior after crossing a temperature around ∼400 K, which is of the order of the critical temperature of the superconducting-like response reported in [15,49].

## 5. Materials and Methods

### 5.1. Samples

In order to investigate the magnetization of graphite samples with different thicknesses, several well-ordered graphite samples were selected, taking into account previous characterization with STEM, XRD, magnetotransport, and particle-induced X-ray emission (PIXE) measurements. The bulk samples were natural graphite samples from Sri Lanka and Brazil [15], as well as HOPG samples of Grade A (ZYA) (Union Carbide, Advanced Ceramic, and SPI) of very high purity [12,34,50]. The total magnetic impurities' concentration of the selected samples was below 7 ppm. See Section 3 in the Supplementary Information for more details on the results of this characterization, as well as the arguments against the speculation that impurities or sample edges are the reason for the observed behavior of the susceptibility.

The embedded interfaces can be well recognized through the STEM picture in Figure 1 (see also [10]) and the existence of the two well-ordered stacking orders by XRD [13,15].

To check for the quality of our samples and the reliability of the experimental setup, we measured the diamagnetic response of bulk samples of thickness $t > 50$ μm with the SQUID and the torque magnetometer. The measured *c*-axis mass susceptibility at 300 K of all bulk samples was $\chi \simeq -(2.2 \pm 0.3) \times 10^{-5}$ emu/g Oe, in good agreement with previously reported values for highly oriented bulk samples [3,30,36,39]. The mass of the graphite samples was measured with a Mettler Toledo AG245 balance. The sizes of the samples with thickness larger than 100 μm were measured with an optical microscope, otherwise using a SEM.

### 5.2. SQUID Measurements

We have used two different magnetometers, a SQUID and a torque magnetometer; the details of this last are given below. The SQUID measurements were done with a SQUID from Quantum Design. The samples were prepared as follows: precharacterized bulk samples were selected, i.e., HOPG ZYA and natural graphite samples from Sri Lanka and Brazil mines. The HOPG ZYA samples were glued with a small amount of cryogenic varnish to a thin silicon substrate ($4 \times 4 \times 0.18$ mm$^3$). In Section 5 of the Supplementary Information, further, we describe the influence of the substrate on the SQUID measurements. The natural graphite samples obtained from a bulk sample were attached to a long, highly pure quartz rod using a small amount of varnish with the *c*-axis of the sample parallel to the applied field. The selected samples were exfoliated with Scotch tape in such a way that the surface of the graphite flake was as flat as possible. The background magnetic signals of the silicon substrate, varnish, as well as of the quartz rod were previously characterized. They were much smaller (absolute value) than the signal of the samples; see Section 5 in the Supplementary Information.

Finally, the graphite samples with their substrates were placed in a plastic straw keeping the magnetic field direction parallel to the *c*-axis of the graphite structure within ±2°. Before each measurement, we left the superconducting solenoid with a remanence below 0.1 Oe, using the oscillating mode option of the SQUID. The magnetization was measured at different fixed fields

at 300 K. Afterwords, the whole measuring process was repeated. All susceptibility values shown and discussed in this manuscript were obtained at fields $H \leqslant 10^4$ Oe.

### 5.3. Torque Measurements

Since graphite is a strong anisotropic material [30], its magnetic properties can be also investigated with a torque magnetometer. The torque measurements were realized using a system that includes a vacuum system with a water-cooled rotating magnet for the generation of fields up to 4 kOe, an AC Resistor Bridge AVS-47 with preamplifier, a Lake Shore Model 325 Cryogenic Temperature Controller, and the torque magnetometer itself. It consists of a piezo-resistive cantilever PRSA-L300 from SCL-Sensor. Tech. Fabrication GmbH. The cantilever has 4 piezoresistors in a Wheatstone bridge circuit, increasing the sensitivity to magnetic moments of the order of $m \sim 5 \times 10^{-10}$ emu at 1 kOe. The Wheatstone bridge is also used to compensate the magnetic field influence on the piezoresistors. Two resistors are placed at the edge of the cantilever for torque measurements and a further two on the cantilever base for current compensation. A picture of the cantilever tips of the two magnetometers with the samples can be seen in Figure 5.

The bulk natural samples were cleaned in ethanol and in an ultrasonic bath for 5–7 min for purification and a further fragmentation of the bulk piece. Ethanol droplets containing graphite flakes dropped onto a silicone substrate covered by a 150-nm silicon nitride layer ($Si_3N_4$). Afterwards, the samples were dried on the substrates for more than one day and were attached at the edge of the cantilever, as shown in Figure 5. Before starting the measurements, we waited for a stable vacuum and temperature (300 K) for two days. Since the diamagnetic response at fields parallel to the *c*-axis of highly-oriented graphite does not depend strongly on temperature (see Figure 2), we restricted the torque measurements to 300 K.

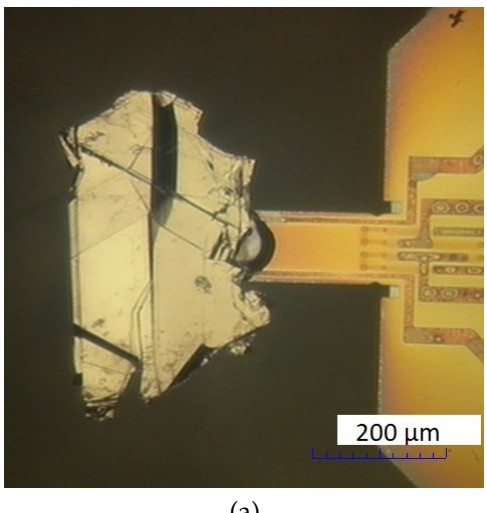
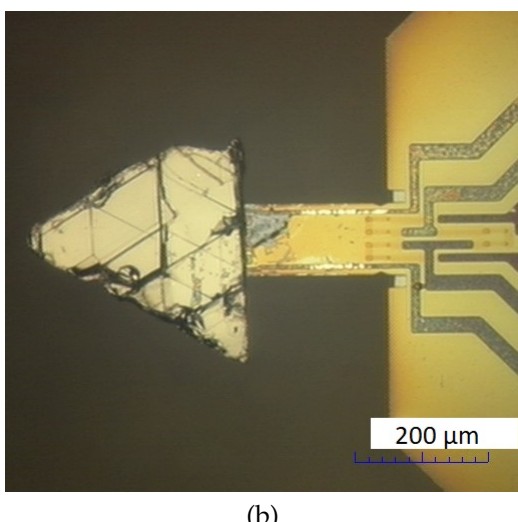

(a)                                                                    (b)

**Figure 5.** Natural graphite flakes on the tips of the cantilevers. The sample thickness was (**a**) 1.2 μm and (**b**) 6.9 μm. The other sample dimensions can be taken from the pictures using the scale bar, which indicates 200 μm. The bend seen in picture (a) does not represent a large portion of the graphite flake, but it comes from a small part just at the sample surface. It represents an insignificant contribution to the total sample mass (or volume) affecting less than 1% the absolute value of the susceptibility. This is estimated through side pictures.

A simple way to check the calibration of the torque magnetometers used is to measure a sample big enough so that one can measure them also with the SQUID. This has been done with one sample. We have checked the calibration through estimates of the strength factors for each cantilever following [51]. After calibration of the torque signal, the susceptibility was calculated by dividing the magnetic moment field slope by the mass of the samples. Further details of the original experimental

data, calibration procedure, and estimates of the spring constant of the cantilevers necessary to calculate the magnetic moment can be read in Section 4 of the Supplementary Information. To check for the magnetic anisotropy and simultaneously the quality of the samples, the angle dependence of the torque was measured using an applied field of 2 kOe and within a range of 270°; see Figure 3.

## 6. Conclusions

Our results show that the absolute value of the diamagnetic susceptibility of highly ordered graphite is not constant, as assumed in the literature, but it depends on the sample even for samples of the same batch and volume. Below a certain thickness, our results indicate a decrease of the absolute value of the diamagnetic susceptibility with the sample thickness. Although more experimental data for thinner samples are needed to assure the observed trend completely, the behavior is compatible with recent studies of the galvanomagnetic properties of graphite. These, as well as our results indicate the existence of a non-uniform heterostructure in real graphite samples strongly affecting their magnetic and electrical properties. The presented results stress the need for a reconsideration of previously-published models used to understand the magnetic and electrical properties of graphite samples.

**Supplementary Materials:** The following Sections and figures in Supplementary Information are available online at https://www.mdpi.com/2312-7481/4/4/52/s1: Section 1, comparison between the thickness dependence of the susceptibility and of the electrical conductivity at 4 K and 300 K with one figure; Section 2, surface to volume ratio with two figures; Section 3, particle induced X-ray emission (PIXE) characterization of the samples impurities; Section 4, original experimental data and the calibration curve with two figures; and Section 5, the influence of the substrate on the SQUID measurements.

**Author Contributions:** B.S. performed all the magnetic measurements and data curation. P.D.E. performed the supervision of the experimental work. B.S. and P.D.E. conceived of and designed the experiments, analyzed the data, and wrote the paper together.

**Funding:** B.S. was supported by the Erasmus Mundus Webb project. P.D.E. was partially supported by the Mincyt from Argentina with the Milstein fellowship during his stay at the Instituto Balseiro in Bariloche, the University of Buenos Aires, and the University of Tucumán, where part of the manuscript was written.

**Acknowledgments:** We acknowledge A.Champi and H. Beth for providing us with the natural graphite samples from Brazil and Sri Lanka and W. Böhlmann for the STEM pictures. B.S. greatly thanks the support of A. Deutschinger, B. C. Camargo, J. Barzola-Quiquia, A. Setzer, V. Zviagin, and M. Stiller.

**Conflicts of Interest:** The authors declare no conflict of interest. The funders had no role in the design of the study; in the collection, analysis, or interpretation of data; in the writing of the manuscript; nor in the decision to publish the results.

## Abbreviations

The following abbreviations are used in this manuscript:

| | |
|---|---|
| 2D | two-dimensional |
| HOPG | highly-oriented pyrolytic graphite |
| SEM | scanning electron microscopy |
| STEM | scanning transmission electron microscopy |
| SQUID | superconducting quantum interference device |

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
