# Peer review of "Diamagnetism of Bulk Graphite Revised"

_magnetochemistry, doi:10.3390/magnetochemistry4040052_

Round 1

Reviewer 1 Report

In the present work, Semenenko and Esquinazi report on the revision of the diamagnetism displayed by different samples of the graphite material. The study was performed both experimentally and theoretically, and having into account parameters such as the temperature dependence of the diamagnetism, the torque angle and the thickness of the studied samples. Overall, the work is OK and the reported information will be of great interest for the readers from the research fields of the magnetochemistry and materials science. So that I recommend the publication in the journal Magnetochemistry as it is.     

Author Response

we thank the Reviewer 1 for the positive report. We have corrected several errors in the written English in the manuscript.  

Reviewer 2 Report

Manuscript ID: magnetochemistry-380767

Referee's Comments on a manuscript (MS) entitled "Diamagnetism of bulk graphite revised" authored by Bogdan Semenenko and Pablo D. Esquinazi.

Reviewers' comments: Bulk graphene materials are attracting significant attention due to their unique electronic and magnetic properties. The diamagnetic response to c axis magnetization in different graphite samples is not been fully understand, even qualitatively. The interpretation of the measured diamagnetism of graphite is mostly based on assumptions that high quality and pure graphite samples are homogeneous, structurally as well as electronically.

In this manuscript authors critically analyze the previous data on internal structures of graphite of different samples including natural graphite crystals and highly oriented pyrolytic graphite. The authors conclude that neither the area nor the density or the electronic characteristics of the interfaces are homogeneously distributed within each bulk sample. This factor causes difficulties in the interpretation of different properties that depend on the magnetic response and conducting 2D interfaces embedded in semiconducting Bernal and/or rhombohedral stacking regions. Authors are convincing by their analyses of measured STEM, SQUID and torque data that the existence of a non-uniform heterostructure in real graphite samples strongly affects the magnetic and electrical properties of graphene. The presented results stress the importance of a revision of previously accepted approaches and suggest a non-intrinsic character of the diamagnetic response in graphene samples for interpretation of the magnetic and electrical properties. The total diamagnetic susceptibility of graphite directly decreases as the sample becomes thinner and below a certain thickness it depends on the internal structure of the graphite sample.

However, authors rule out the possibility that decrease of the sample thickness can decrease in the total diamagnetic susceptibility due to the possible increase of the paramagnetic or ferromagnetic contribution or other factors. The bulk Curie paramagnetism unobservable in bulk graphite materials which are diamagnetic, can exhibit paramagnetic or even ferromagnetic behavior in pyrolytic nanographite that can be attributed also to the other factors such as localized spins at the edge states (IEEE Transact. Magn. 52 3 2016) which seems absent in microcrystalline graphites. The studies (Carbon 44 1225 2006) show that during continuous annealing the nanoparticle structure changes from more defective to less defective. A detailed analyzes of the changes at the nanoscale in the orbital diamagnetic susceptibility and localized spin concentration when the sp3 /sp2 -carbon ratio was varied by varying the annealing temperature.

Referee's recommendation: This manuscript is rather interesting and definitely provides new interpretation regarding diamagnetic origin of magnetic response by showing its strong depends on inhomogeneous structure of different samples. Authors found that the diamagnetic susceptibility decreases with the sample thickness and show that the existing paradigm in interpretation of results is misleading since the experimental magnetic and electrical data of bulk graphite are assumed have homogeneous structures. Moreover, the authors found that the absolute value of the diamagnetic susceptibility of highly ordered graphite is not constant, as assumed in the literature, but it depends on the structural details even for samples of the same batch and volume. These variations of the susceptibility for similar samples of different thickness suggest a non-intrinsic origin of the main diamagnetic signal. Below a certain thickness, the results indicate a decrease of the absolute value of the diamagnetic susceptibility with the thickness. The graphene samples are inhomogeneous in agreement with scanning transmission electron microscopy images and X-ray diffraction data. In this manuscript the wide range of composition analysis and data are used to study the electronic and magnetic properties.

In overall, manuscript with many interesting results is well written and provides a useful contribution to the field. However, there are several misspellings (e.g. "mayor role”, “comercial”). By taking into account the above comments, I would ask to make a spelling check and resolve these MINOR/ TYPOGRAPHICAL ISSUES. Thus, I can recommend publish this slightly revised manuscript in Magnetochemistry journal by reflection of the annealing effect and corresponding spin edge effects and other factors mentioned in above relevant publications on the subject.

Author Response

We thank Reviewer 2 for the positive report and for providing us with useful references.

Apart from corrections in the written English overall in the main manuscript and in the supplementary information, we have included two new paragraphs and two new citations.

The reviewer 2 wrote:

"However, authors rule out the possibility that decrease of the sample thickness can decrease in the total diamagnetic susceptibility due to the possible increase of the paramagnetic or ferromagnetic contribution or other factors. The bulk Curie paramagnetism unobservable in bulk graphite materials which are diamagnetic, can exhibit paramagnetic or even ferromagnetic behavior in pyrolytic nanographite that can be attributed also to the other factors such as localized spins at the edge states (IEEE Transact. Magn. 52 3 2016) which seems absent in microcrystalline graphites. The studies (Carbon 44 1225 2006) show that during continuous annealing the nanoparticle structure changes from more defective to less defective. A detailed analyzes of the changes at the nanoscale in the orbital diamagnetic susceptibility and localized spin concentration when the sp3 /sp2 -carbon ratio was varied by varying the annealing temperature."

From line 205 we included the following sentences:

"The influence of localized spins at the graphene edges (or the presence of nitrogen impurity atoms) [42] on the total susceptibility of our graphite samples should be negligible, since the lateral surface of the samples is much smaller than their volume, and the magnitude of the total susceptibility does not correlate with the lateral surface of the samples (see Appendix B in supplementary information). Moreover, our magnetization measurements were carried out at room temperature, which strongly diminishes any ferromagnetism or paramagnetism influence due to edge states or other possible localized magnetic moments."

42. Sharoyan, E.; Mirzakhanyan, A.; Gyulasaryan, H.; Sanchez, C.; Kocharian, A.; Bernal, O.; Manukyan, A. Ferromagnetism of Nanographite Structures in Carbon Microspheres. IEEE Transactions on Magnetics 2016, 52, 1–3. doi:10.1109/TMAG.2016.2527792.

From line 236 we included following sentences:

"The paramagnetic Pauli susceptibility of free carriers may compensate the diamagnetic contribution of the core susceptibility [43]. Also, those contributions are not expected to change with sample size. Therefore, we do not take them into account to estimate the total diamagnetic susceptibility of graphite."

43. Osipov, V.; Enoki, T.; Takai, K.; Takahara, K.; Endo, M.; Hayashi, T.; Hishiyama, Y.; Kaburagi, Y.; Vul’, A. Magnetic and high resolution TEM studies of nanographite derived from nanodiamond. Carbon 2006, 44, 1225 – 1234. doi:https://doi.org/10.1016/j.carbon.2005.10.047